# Low Glycemic Index Biscuits Enriched with Beetroot Powder as a Source of Betaine and Mineral Nutrients

**DOI:** 10.3390/foods14050814

**Published:** 2025-02-27

**Authors:** Jasmina Mitrevski, Nebojša Đ. Pantelić, Jovanka Laličić-Petronijević, Jovana S. Kojić, Snežana Zlatanović, Stanislava Gorjanović, Stevan Avramov, Margarita S. Dodevska, Vesna V. Antić

**Affiliations:** 1Faculty of Agriculture, University of Belgrade, Nemanjina 6, 11080 Belgrade, Serbia; prodaja@zdravahranaolea.com (J.M.); jovankal@agrif.bg.ac.rs (J.L.-P.); 2BioSense Institute, University of Novi Sad, Dr Zorana Djindjica 1, 21000 Novi Sad, Serbia; jovana.kojic@biosense.rs; 3Institute of General and Physical Chemistry, Studentski Trg 12-16, 11000 Belgrade, Serbia; snezana.zlatanovic@gmail.com (S.Z.); stasago@yahoo.co.uk (S.G.); 4Institute for Biological Research “Sinisa Stankovic”, National Institute of Republic of Serbia, University of Belgrade, Bulevar Despota Stefana 142, 11060 Belgrade, Serbia; stevan@ibiss.bg.ac.rs; 5Institute of Public Health of Serbia “Dr Milan Jovanović Batut”, Center for Hygiene and Human Ecology, Dr Subotica 5, 11000 Belgrade, Serbia; margarita_dodevska@batut.org.rs

**Keywords:** biscuits, beetroot, betaine, minerals, glycemic index, sensory analysis

## Abstract

This study aimed to evaluate the potential of beetroot powder (BP) as a functional ingredient in biscuits by investigating its effects on nutritional composition, sensory properties, and glycemic response. The primary goal was to determine whether BP could serve as a natural alternative to synthetic additives while maintaining product stability and consumer acceptability. Biscuits were formulated by replacing spelt flour with 15, 20, and 25% BP. The functional impact of the BP was assessed based on betaine content, macro- and microelements, glycemic index (GI), and acrylamide concentration. Thermal analysis (DSC and TGA) and water activity measurements confirmed the BP’s stability during six months of storage. Increased BP content led to higher betaine levels and mineral enrichment, particularly with potassium and phosphorus among the macroelements and zinc among the microelements. Sensory analysis identified biscuits with 20% BP as the most preferred, maintaining acceptable ratings even after six months. Hardness initially increased with BP incorporation but decreased over time (*p* < 0.05). The acrylamide content in the BP-enriched biscuits was significantly lower than in control samples and well below the reference safety threshold. Notably, consuming beetroot biscuits did not trigger a sharp postprandial glucose spike, with the GI of the most acceptable sample (20% BP) measured at 49 ± 11. These findings confirm that BP improves the nutritional and sensory characteristics of biscuits while ensuring product safety and stability, supporting its application as a natural functional ingredient in confectionery products.

## 1. Introduction

The science and the food industry trends are increasingly focused to functional food with special health benefits. As a result, there is a growing interest in plant-based ingredients that contribute to maintaining health and protecting against disease. Beetroot (*Beta vulgaris* L.) is a biennial, herbaceous plant whose roots and leaves are a valuable source of dietary fiber, antioxidants, minerals, and vitamins in human nutrition [1,2,3]. Beetroot is commonly utilized as a dietary source for human consumption. It should be stored under controlled conditions to maintain its quality for three to five months. For long-term preservation of nutritional properties, fresh beetroots can be dehydrated at low temperatures, ground into powder, and then stored in tightly closed containers. The powder, or beetroot powder, can be added to various food products to increase their nutritional and health-promoting properties [4,5,6].

Nutritionists often perceive flour-based confectionery products, such as biscuits, as high in sugar, salt, and fat and therefore classify them as “empty calories”. On the other hand, this type of product enjoys great popularity among consumers of all ages, especially children, whose consumption is steadily increasing. Therefore, enrichment of confectionery products with bioactive components can be considered part of a nutrition improvement strategy. Furthermore, popular foods such as confectionary products represent an effective means of incorporating functional ingredients and are therefore targeted by the growing market for managing health disorders. It is undeniable that functional food with high fiber and bioactive ingredient contents contributes significantly to the proper functioning of the entire organism, accelerates metabolism, and improves digestion. This type of food can reduce obesity and prevent or delay the onset of many diseases, such as cardiovascular disease, diabetes, and certain types of cancer [7,8]. Biscuits are an ideal type of product for incorporating functional ingredients due to the easy availability of nutrients. Biscuits can be enriched with mineral and vitamin complexes or nutrients that are especially suitable for babies, children, elderly consumers, and people with special needs such as obesity or diabetes. Numerous scientific works have been published on producing different types of biscuits with functional ingredients. This has been particularly influenced by the trend of replacing wheat flour, the traditional primary raw material in biscuits, with flour from other cereals, pseudo-cereals, or non-cereal flours, such as flour from stone fruits, tropical fruits, oil cakes, tuberous plants, as well as various by-products of the food industry. So far, studies have been published showing the impact of using different fruit-based ingredients such as blueberry and blackberry [9], apple pomace [10,11], citrus peel [12], mango peel [13], and carrot [14] on the antioxidant and nutritional characteristics of biscuits. Beetroot is also used in various forms as an ingredient in confectionery products. In addition to the raw form, juice, powder, extracts, and processing residues (pomace) have also been used as a source of bioactive compounds from beets [3,15]. The results of the study by Ingle et al. [16] show that biscuits with added dried beet leaf powder (up to 12%) contain a significant amount of protein, fiber, minerals, plant pigments, omega-3 fatty acids, and vitamins. According to Krejcova et al. [17], dried beetroot can be consumed as a healthy snack. The nutrients in beetroot and its powder include carbohydrates, dietary fiber, protein, and a small amount of fat, along with essential minerals such as calcium, potassium, zinc, sodium, magnesium, and iron. It is also a rich source of vitamins *C*, *A*, and *B*_6_, niacin, and folic acid, as well as bioactive compounds such as betalains (betacyanins and betaxanthins) and polyphenols, which contribute to its antioxidant and health-promoting properties [18,19,20]. In addition, regular consumption of beetroot can reduce symptoms of hypertension, type 2 diabetes, dementia, arteriosclerosis, and kidney disease [2,21]. Kohajdova et al. [22] determined that adding beetroots made roll dough more stable and improved water absorption.

Our previous work presented the physico-chemical, nutritional, microbiological, and functional properties of biscuits made from spelt flour with different proportions of BP [23]. Beetroot powder was added to improve the antioxidant properties and increase dietary fiber content. Two series of biscuit samples were prepared at various temperatures (150 and 170 °C), with BP replacing up to 50% of the spelt flour, so that the percentage of BP in the baking mixture (dough), was 15, 20, and 25 wt %. The physicochemical and functional properties of beetroot biscuits (BPB) were evaluated during 6 months of storage. It was found that the value of water activity (aw) was from 0.35 to 0.56, indicating appropriate storability. The highest concentration of betalains, polyphenolics, and fiber (7.60%) was found in the biscuits with the maximal content of BP incorporated. Good retention of betalains, polyphenolics, and flavonoids during storage was confirmed, while the antioxidant activity (DPPH, FRAP) decreased slightly over time. Additionally, there was almost no effect of baking temperature on antioxidant activity. Building upon these findings, the present study aims to further investigate the functional potential of BP in biscuits by assessing its role as a natural alternative to synthetic additives, particularly in terms of glycemic response, product stability, and safety. The main focus is on whether BP can enhance the nutritional quality and sensory appeal of biscuits while maintaining acceptable technological properties. Furthermore, this study provides novel insights into the effect of BP on the glycemic index of biscuits and its influence on acrylamide formation, aspects that have not been previously investigated. Firstly, a thermal analysis of BP was performed to test its prolonged stability and the possibility of broader application in the production of confectionery products. Afterwards, the betaine content, hardness, and sensory properties of the biscuits were examined shortly after production and after 3 and 6 months of storage. The content of micro- and macro elements was established immediately after baking. The influence of the addition of BP on the glycemic index was determined on the most sensory-acceptable biscuit sample. The impact of baking temperature and BP content on the acrylamide concentration in the biscuits was also examined, providing valuable insights into product safety.

## 2. Materials and Methods

### 2.1. Materials

Preparation of BP and biscuits with 15, 20, and 25 wt % BP in the dough is described in detail in Mitrevski et al. [23]. Detroit variety beetroot was thoroughly washed, peeled, and sliced into 1 mm thick rings. The slices were arranged on drying trays without overlapping and dehydrated at 52 °C for 24 h until a constant mass was achieved. After cooling for 3 h, the dried beetroot was ground using a high-power grain grinder (VITA-MIX CORP, 1200 W, Olmsted, OH, USA). Biscuits were made by substituting a portion of wholemeal spelt flour (SF) with beetroot powder (BP).

Two batches were prepared and baked at 150 °C and 170 °C, with a control batch made without beetroot powder. Each type of biscuit was placed on a separate tray, with a total of 20 biscuits per tray, and baked in a multi-stage oven. After baking, they were left to cool on the tray for 2 h. Sample designation is provided in Table 1.

The beetroot powder and biscuits were stored in sealed PVC containers, protected from light and moisture, and kept until analysis at 3 and 6 months. All analyses described below were performed in triplicate.

### 2.2. Differential Scanning Calorimetry (DSC)

A differential scanning calorimeter (DSC, Q1000, TA Instruments, New Castle, DE, USA) was used to perform DSC experiments. Three beetroot powder samples (5–7 mg) were taken and placed into aluminum pans with pinhole lids, cooled from 20 to −90 °C, then equilibrated for 5 min and scanned initially from –90 to 150 °C, with a controlled heating rate of 5 °C/min under a N_2_ purge flow of 50 mL/min. Each thermogram was analyzed using TA Advantage Universal Analysis 2000 software, version 4.5A. Glass transition was characterized by onset temperature (T_gon_), midpoint temperature (T_g_), and end temperature (T_gend_) of the heat capacity change (ΔCp). The DSC Q1000 with Advanced TzeroTM technology provides a direct measure of heat capacity with a single scan.

### 2.3. Thermogravimetric Analysis (TGA)

The thermal degradation of the beetroot powder samples was studied using thermogravimetric (TG) analysis. Measurements were performed using a TA Instruments TGA Q500 Thermogravimetric Analyzer (Dover, DE, USA) under a 60 mL/min nitrogen flow. Three independent samples (5–7 mg) were placed into the platinum crucible, loaded into the TG furnace, and heated from 25 to 700 °C. The obtained TG curves and derivative curves (dTG) were used to analyze the thermal properties of the samples. The characteristic temperatures of the thermal decomposition process (start temperature—T_s_ and end temperature—T_end_) were determined from the dTG curve using TA Universal Analysis 2000 software. T_s_ and T_end_ were determined as a minimum at the beginning and end of thermal decomposition peaks on the dTG curve.

### 2.4. Water Activity (aw) and Moisture Content

Moisture was determined gravimetrically at 105 °C in a temperature-controlled oven to a constant weight. The water activity of the BP and BPB samples was determined using LabSwift-aw, Novasina AG, Lachen, Switzerland, at 25 °C.

### 2.5. Hardness

The hardness of the biscuits was determined based on diametrical compression using a TA-XT2 Texture Analyzer (Stable Micro Systems, Surrey, Great Britain). The measurements were performed according to the method described by Belović et al. [24]. In this approach, the instrument was equipped with a flat blade (HDP/BS) and a 30 kg load cell. Hardness represents the force at which the biscuit’s first break occurs and the maximum of the obtained force dependence curve over time. The hardness of the sample is expressed as the mean value in kilograms. The instrument setting parameters during the test were as follows: speed before measurement—1.5 mm/s, speed during measurement—3 mm/s, speed after measurement—10 mm/s, and trigger force—25 g.

### 2.6. Determination of Betaine

The extraction and measurements were performed according to the method describe by Kojić et al. [25]. Briefly, 2 g of previously ground sample was weighed and suspended in 25 mL of methanol. The sample was homogenized on a vortex for 10 min and then extracted for half an hour in an ultrasonic bath. After that, the cuvette with the sample was centrifuged for 10 min at 5000 rpm. A total of 3 mL of the upper methanol layer was separated and evaporated to dryness. The residue was redissolved in 2 mL of water and filtered through a membrane filter (regenerated cellulose, 0.22 mm pore size, 25 mm diameter, Agilent Technologies, Santa Clara, CA, USA). Quantitative determination of betaine was performed using an Agilent HPLC system chromatograph, USA, equipped with a Kinetex^®^HILIC column (Phenomenex, Aschaffenburg, Germany) with dimensions of 2.6 μm, 100 × 2.1 mm and an ELSD detector (1290 Infinity ELSD, Agilent Technologies, USA). The mobile phase consisted of acetonitrile solution and 10 mM acetate buffer, pH 3.7 (72.28% acetonitrile and 27.72% buffer). The mobile phase flow rate was 0.5 mL/min. An isocratic mode of operation was used, with a total determination time of 10 min and an injected volume of 5 μL. The injector was at room temperature. The detector was set according to the following parameters: evaporator temperature—40 °C, atomizer temperature—55 °C, gas flow rate—1.60 L/min. The calibration standard was anhydrous betaine with a purity of 98%. All analyses were performed in duplicate.

### 2.7. Determination of Macro- and Micro Elements

The samples were mineralized using a microwave digester Advanced Microwave Digestion System (ETHOS 1, Milestone, Milano, Italy). A segmental high-pressure rotor, HPR-1000/10S, was used. The samples were directly measured in quartz inserts. After measurement, oxidizing reagents were added to the samples: 4.5 mL HNO_3_ (65 wt %, Suprapur^®^, Merck KGaA, Darmstadt, Germany), ultrapure nitric acid, hydrogen peroxide, 0.5 mL H_2_O_2_ (30 wt %, Suprapur^®^, Merck KGaA, Darmstadt, Germany). After filling, the cuvettes were placed in the holder and closed according to the manufacturer’s protocol. Digestion was performed for 20 min at 180 °C at a pressure of 100 bar. The solution was then cooled to room temperature and diluted in a volumetric flask to a fixed volume of 25 mL. Dilution was performed using ultrapure water with 0.05 μS/cm of electrical conductivity.

Induced coupled plasma with optical emission spectrometry (ICP-OES), Thermo Scientific iCAP 6500 Duo ICP (Thermo Fisher Scientific, Cambridge, UK) was used for the measurement of the concentrations of micro-, macro-, and toxic elements in the prepared dilutions. The sample was introduced into the plasma via direct aspiration of the liquids. Standard solutions for the calibration instruments were prepared from certified solutions: SS-Low Level Elements ICV Stock and ILM 05.2 ICS Stock 1 (both from VHG Labs, Inc-Part of LGC Standards, Manchester, NH 03103, USA), SS-Low Level Elements ICV Stock and ILM 05.2 ICS Stock 1 (both from VHG Labs, Inc- Part of LGC Standards, Manchester, NH 03103, USA). The standard solutions were in the range of 1–50,000 μg/L. The correlation coefficient was >0.99 for all elements. For quantification, the wavelength of light that had the best standard–sample match according to all spectrophotometric criteria was taken. Each sample was measured three times (n = 3). The relative standard deviation was RSD < 1%. The limit of detection (LOD) for all elements was 0.05–1.5 μg/L, and the limit of quantification (LOQ) was from 0.1 μg/L to 5 μg/L. The measured masses of the samples, concentrations of elements in the solution, and dilutions gave the final concentrations of elements in the samples in mg/kg sample (ppm). An analytical quality control process was performed using certified reference material (CRM): fish protein for trace metals DORM 4 (NRCC, National Research Council Canada, Ottawa, ON, Canada) and EPA Method 200.7 LPC Solution (ULTRA Scientific, North Kingstown, RI, USA). A match in the range of 98 to 103% with the certified value (recovery) was achieved for all certified elements.

### 2.8. Determination of Acrylamide

The acrylamide (AA) content was determined using quantitative GC-MS analysis of the brominated derivative. Derivatization of AA resulted in a nonpolar 2,3-dibromopropanamide, which was less volatile than acrylamide. By further transforming 2,3-dibromopropanamide into 2-bromo-2-propenamide, the potential risk of dehydrobomination in the injector or GC column, which can significantly affect the sensitivity and selectivity of the method, was avoided. The internal standard (methacrylamide) was derivatized by bromination into 2,3-dibromo-2-methylpropanamide, the final product, given that it cannot undergo the dehydrobromination reaction. Spike experiments gave 94.8 to 102.7% recoveries, with relative standard deviations of 3.8% to 10.6%. The LOD and LOQ values of the validated method were 5 μg/kg and 25 μg/kg, respectively. The acrylamide extraction, bromination, calibration, and quantification procedures are described in detail in the technical specification of FprCEN/TS (2017).

### 2.9. Sensory Analysis

Sensory evaluation of the biscuit samples with the addition of beetroot was carried out at the Faculty of Agriculture, University of Belgrade with the participation of experienced tasters (8 evaluators who met the criteria of standard ISO 8586:2023 [26]). The following sensory properties were evaluated: appearance (color, surface, size, and shape), texture (structure, bakedness, breakage, chewiness), smell, and taste. The overall sensory quality was determined using the scoring method. A point scale from 1 to 5 was used for the evaluation, with the possibility of assigning half and quarter points. An importance coefficient (IC) was determined for each attribute of sensory quality according to the individual influence the evaluated parameters on the overall quality so that the sum of the importance coefficients was 20. By adding the individual grades, a complex indicator was obtained that represents the total sensory quality, expressed as a % of the maximum quality. Dividing this value by the sum of the importance coefficients gives the weighted mean value of the grade, representing the overall sensory quality of the examined biscuits. The biscuit samples were classified into quality categories based on the weighted mean value of their grade: excellent, 4.50–5.00; very good, 3.50–4.50; good, 2.50–3.50; unsatisfactory, <2.50.

### 2.10. Determination of Glycemic Index

For this analysis, the biscuit with the best sensory evaluation was chosen, sample **C2**, with 20% beetroot powder. The glycemic index was determined according to the internationally recognized ISO 26 642:2010 standard [27]. The Ethical Commission of the Institute for Public Health of Serbia, “Dr. Milan Jovanović Batut,” approved this study and issued an Ethical permit for its implementation under serial number 5464/1. This study included 10 healthy adults aged 22–59 years with a body mass index (BMI) range from 19 kg/m^2^ to 23 kg/m^2^ and fasting glucose level ≤ 6 mmol/L. Informed consent was obtained from all the respondents. One test was performed per week. The participants fasted for at least 12 h and avoided intense physical activity, smoking, and alcohol consumption. Blood samples were taken 5 min before consumption (0’). Fasting glucose tests were performed with 25.0 g of glucose dissolved in 250 mL of water and consumed within 5 min or with an amount of beetroot biscuit containing 25.0 g of available carbohydrates. Blood was sampled at 15’, 30’, 45’, 60’, 90’, and 120’ min after consumption. A graph of changes in glucose concentration as a function of time was made, and the area under the curve (AUC) was calculated. The glycemic index, i.e., the ratio of postprandial blood glucose after consuming a biscuit containing 25.0 g of available carbohydrates to 25.0 g of reference carbohydrate (glucose), was calculated according to the formula GI = AUC/AUCref × 100. The AUC and AUCref are the areas under the curve of the time dependence of glucose concentration in capillary blood measured after consuming the biscuit sample containing 25.0 g of carbohydrates or a solution of 25.0 g of pure glucose. Foods that quickly absorb carbohydrates have a high GI (GI ≥ 70). A GI of 56 to 69 is medium, and a GI ≤ 55 is low. Glycemic load (GL) is the product of GI and total available carbohydrates in a given serving of food divided by 100. GL values are also categorized as low (≤10), medium (>10 to <20), or high (≥20).

### 2.11. Statistical Analysis

The results obtained in this work were analyzed using the IBM SPSS Statistics 25 program (IBM, Armonk, New York, NY, USA). The sample size is shown, and descriptive statistics are calculated (mean value, median, minimum, and maximum). Measures of variability and standard deviation were calculated for the corresponding groups of samples. The Kruskal–Wallis and the Wilcoxon–Mann–Whitney test (a non-parametric analog of the independent samples *t*-test) were used to assess the significance of differences between the properties measured in the biscuits during a six-month storage period (immediately after production, after three months, and after six months) and between biscuits with four different beet powder contents (0%, 15%, 20%, 25%). The test *p*-values are shown for each measured characteristic. The significance level was set to 0.05. The complete sample on which the analyses were performed consisted of 50 replicates.

## 3. Results and Discussion

Beetroot, as well as its juice, powder, pomace, and extracts, are considered excellent raw materials that can serve as a source of bioactive compounds during the production of functional food, i.e., as functional ingredients. Due to the significant content of betalains, phenolic compounds, and inorganic ingredients, beetroot-based ingredients increase the final product’s shelf life and reduce the use of synthetic additives. The current study delves into the thermal properties of beetroot powder, using techniques such as DSC and TGA for a comprehensive understanding. Furthermore, the impact of BP incorporation on various functional properties, including betaine content and macro- and microelements, and the glycemic index and load of the biscuits were explored. The influence of the beetroot incorporation on acrylamide content in the biscuits was also examined. In addition, a sensory acceptability test was conducted to assess the potential consumers’ response to the beetroot biscuits. This thorough investigation provides a reliable and detailed overview of the possible benefits and implications of using beetroot in functional food production.

### 3.1. Thermal Behavior of Beetroot Powder—DSC and TGA Results

A thermal analysis using DSC and TGA techniques was performed to examine in more detail the stability of BP and the possibility of its application in the production of confectionery products. The DSC thermograms of BP, recorded immediately after drying in a laboratory dehydrator at 52 °C and after 6 months of storage at room temperature, are shown in Figure 1. The results of the DSC analysis are given in Table 2.

Both DSC curves of BP show a stepwise change, indicating the existence of glass transition temperature at 41.0 °C at the beginning of storage and 37.4 °C after 6 months of storage. At the same time, the water activity was 0.38 and 0.36, respectively. The obtained results are comparable to previously published data related to beetroot powder, where the glass transition temperature was 50.6 °C [28], as well as with the results for apple pomace flour, which had T_g_ values between 28 and 38 °C and a low water activity of 0.2–0.4 [29].

Figure 1 shows that the glass transition follows a broad endothermic peak in the temperature range from 132.5 to 186.7 °C (beginning of storage) and 134.8 to 185.0 °C (after 6 months of storage). This peak is due to the melting of ordered crystal structures, most likely mono- and disaccharides [30]. BP is a multicomponent system in which components such as starch, sugars, proteins, and color can be partially amorphous and crystalline. They can also be mixable or non-mixable, leading to a multiphase microstructure [31]. The thermal behavior of BP indicates the existence of this multiphase structure containing a partially amorphous phase (showing glass transition) and a partially crystalline phase of BP (showing endothermic melting). A single glass transition temperature indicates good mixing of the components in the BP. At temperatures above 220 °C, the endothermic transition turns into an exothermic one, marking the beginning of sample degradation. Low values of aw (below 0.4) and T_g_ values above average storage temperatures ensure prolonged stability of the BP. Similar results were obtained by Zlatanović et al. [29] when examining flour samples from apple pomace. The glass transition temperature of the BP was correlated with moisture content and aw (Table 2). It is known that the T_g_ of biological materials and foods decreases with increasing water content, which acts as a plasticizer [32]. From the results obtained in this work, it can be concluded that smaller amounts of stable BP can be obtained via drying in a dehydrator and that the obtained TGA thermograms correspond to the complex composition of BP, with a high proportion of total carbohydrates (57.06 ± 1.20), dietary fibers (19.90 ± 0.55), and protein (11.4 ± 0.20), which was shown in our previous publication [23].

The thermal stability of the BP was examined via thermogravimetric analysis in the interval from 25 to 700 °C, immediately after dehydration and after 6 months of storage at room temperature. The TG and dTG curves in Figure 2 show several stages of mass loss up to 700 °C. The dTG curve provides information on the rate of change in mass loss when the BP was heated. The initial and final temperature of each observed stage (T_s_ and T_end_, respectively) and the percentage of mass loss in a particular degradation stage are shown in Table 3. At lower temperatures, evaporation of the water remaining after drying occurred. The first and second stages were observed from 29 to about 130 °C and from 130 to about 180 °C, respectively, corresponding to less than 4% mass losses. The total mass loss up to 180 °C was less than 7.5% and corresponded to the evaporation of adsorbed water and readily volatile components. The third and main stage of degradation, which took place in the temperature interval from 180 to about 270 °C, corresponded to a mass loss of about 40%, and it occurred due to the decomposition of organic compounds of smaller molecular weights that entered the BP composition, such as mono- and disaccharides. It is also known from the literature that the degradation of glucose, fructose, and sucrose takes place in this mentioned temperature interval [30]. The maximum rate of change in mass loss corresponded to a temperature of about 210 °C (maximum on the dTG curve). The fourth degradation stage, which occurred between 270 and 700 °C, refers to the decomposition of high-molecular-weight products that are formed by polymerization reactions of products from the third degradation stage or are already present in the sample (lignocellulosic components).

The fourth stage corresponds to a mass loss of about 30%, which indicates that the proportion of lignocellulosic compounds (hemicellulose, cellulose, and lignin) is significant, as they are degraded in the mentioned temperature range [33,34,35,36]. The rest at 700 °C was 25% of the initial mass of the BP. The obtained results confirmed the stability of the BP at the baking temperature, which enables its broader application in the production of confectionery products.

### 3.2. Hardness of the Biscuits

The chemical composition most significantly affects the hardness, one of the critical parameters of biscuits. The hardness should be high enough to prevent the biscuits from breaking during transport and, on the other hand, low enough to be suitable for consumption. The acceptable hardness of biscuits made from refined flour and the optimal amount of fat and sugar should be around 2000–3000 g [37]. Published research has shown that the hardness increases with increasing proportions of wheat flour due to the properties of gluten. Variations in hardness may occur due to lipid, protein, and starch contents [38]. Research has shown that adding dietary fiber from fruit significantly increases the hardness of biscuits [39], and similar results have been obtained when using vegetable fiber [40]. Beetroot powder had a decisive influence on the hardness, as biscuits with beetroot had higher hardness values than the samples without beetroot. The higher hardness of the tested biscuits was also associated with large bran particles originating from the spelt flour, which was the basis for the biscuit’s production. The results, showing an increase in the hardness with an increase in the proportion of BP, are presented in Table 4. Compared to the biscuits made from spelt flour, with hardness values of 8.98 ± 1.36 g and 6.63 ± 0.90 g, our results are promising for the samples labeled **B1** and **D1**, even after 6 months of storage, in terms of hardness measurements [41]. However, all of the samples showed a significant decrease in hardness (*p* < 0.05) during the storage period of six months. The samples with 20% beetroot powder, **C1**, baked at a lower temperature, showed the most significant decrease, by 58% in the sixth month, while sample **D2**, with 25% beetroot powder and baked at higher temperature, showed a reduction of 50%. As moisture loss occurs during storage, variations in hardness over six months may be due to changes in lipid, protein, and starch contents, affecting the water distribution in the biscuits. 

### 3.3. Betaine Content

Betaine is a non-essential nutrient whose primary physiological function is donating a methyl group in transmethylation reactions and protecting cells from osmotic stress [42]. Since it cannot be formed endogenously, it must be obtained through food. Dietary betaine intake reduces metabolic syndrome and lowers total homocysteine content and the risk of cardiovascular disease [43,44]. Betaine can also reduce the risk of developing cancer, liver disease, depression, and peripheral neuropathy [45,46]. Due to its significant impact on health, the betaine content in biscuits has been the subject of many studies. Within the list of permitted health claims, the EU Commission Regulation No. 432/2012 states that food containing at least 500 mg of betaine per quantified portion contributes to the normal metabolism of homocysteine, provided that the total daily intake of betaine reaches 1.5 g. Research by Ross et al. [47] has shown that cereals are the primary source of betaine in the Western diet, and the importance of using wheat, especially whole grain products, to meet daily betaine requirements has been highlighted. In research conducted by Kojić et al. [25], it was shown that spelt grain and whole spelt flour contain more betaine than ordinary wheat flour (56.52–81.46 mg/100 g, 125.640 mg/100 g and 31.00 mg/g, respectively). Spelt-based cookies enriched with wild garlic leaves osmodehydrated in sugar beet molasses proved to be superior in terms of betaine content, with 1.56 times higher amounts compared to control cookies [48].

The betaine content in the biscuits with BP in the present study is shown in Table 5. Considering the high content of betaine in the BP in this work, from 898.95 to 1197.32 mg/100 g (*p* < 0.05), it was expected that the analyzed biscuits would have a high betaine content. Moreover, it was found that with an increase in the content of beetroot powder, the betaine content also increased proportionally. Samples with 25% BP showed the highest betaine content, 307.34–403.16 mg/100 g (**D1**) and 289.69–387.28 mg/100 g (**D2**). The betaine content was higher in the sixth month than at the beginning of storage. The samples with 15 and 20% BP showed the same trend of increasing betaine content during storage: 258.82–329.68 mg/100 g (**B1**), 179.85–196.15 mg/100 g (**B2**), 283.13–375.22 mg/100 g (**C1**), and 276.12–365.69 mg/100 g (**C2**), as shown in Table 5. Baking temperature had no influence on the betaine content in the samples with 20 and 25% BP. Sample **B2,** with 15% BP baked at 170 °C, had significantly less betaine content (*p* < 0.05) compared to **B1**, baked at 150 °C, across all the storage periods. Considering that food containing at least 500 mg of betaine per quantified portion contributes to normal homocysteine metabolism, it can be concluded that biscuits with 20 and 25% BP and with approximately 300 mg of betaine per 100 g (at the beginning of storage) can be classified as food that contributes to improving consumers’ health status. The total betaine content was increased over four times in the biscuits with the addition of 20–25% beetroot powder.

### 3.4. Macro- and Microelement Contents

Macro- and microelements are very important for a healthy and balanced human diet and are essential for the functioning and maintenance of cells, tissues, organs, and the entire human organism. Insufficient intake leads to a deficit, while excessive intake leads to toxic effects. Between these two extremes lies the amount expressed as the recommended intake that prevents deficits and ensures quality functioning of the organism. Recommended intake depends on population, gender, age, and lifespan.

The importance of macro- and microelements for plant metabolism is the same as for humans, but their physiological role in plants differs. Many studies on micro- and macroelements have shown that their contents in plant materials is high [49,50,51]. Thus, this work found that beetroot, as well as products with beetroot powder, contain a significant amount of different micro- and macroelements. The following macroelements were identified: sodium (Na), potassium (K), magnesium (Mg), calcium (Ca), phosphorus (P), and sulfur (S), as well as microelements and trace elements including zinc (Zn), manganese (Mn), iron (Fe), copper (Cu), selenium (Se), boron (B), chromium (Cr), cobalt (Co), aluminum (Al), arsenic (As), barium (Ba), cadmium (Cd), lithium (Li), nickel (Ni), strontium (Sr), and lead (Pb) based on ICP-OES analysis. The contents of macro- and microelements in the BP, spelt flour, and biscuit samples are shown in Table 6 and Table 7. Sodium is needed to conduct nerve impulses and maintain water balance in the body [52]. The sodium concentration was 2733 mg/kg in the BP, while the concentration in the biscuit samples increased statistically significantly from 564 to 1220 mg/kg, in proportion to the increase in BP content. The content of potassium, which comes predominantly from beetroot, in which its concentration was 7180 mg/kg, increased with increasing BP content and ranged from 1059 mg/kg to 2743 mg/kg in the biscuits. A statistically significant increase was observed between the sample without BP and the samples with 15 and 20% BP. Increasing the BP proportion from 20 to 25% showed no statistically significant difference in potassium content. A similar amount of potassium was found in apple pomace flour (4228–6398 mg/kg) [11]. Since potassium helps maintain the body’s acid–base balance and reduces the risk of cardiovascular disease and stroke, and given that the daily potassium requirement of an adult about 4700 mg/day, BP can be considered an excellent source of potassium. Compared to other sources of potassium, BP contains up to seven times more potassium than rice flour (974 mg/kg) and about five times more than wheat flour (1500 mg/kg) [53]. Magnesium is important as a coenzyme for numerous enzymes and therefore for human metabolism. In the studied samples, there were no significant differences in the magnesium content between the analyzed biscuit samples baked at both applied temperatures (Table 6). The highest calcium content of 270 mg/kg was found in the sample with 25% BP. Compared to the control sample, the calcium content statistically significantly increased with increased BP content in the biscuit samples. Calcium is an essential mineral that prevents the development of osteoporosis, improves the function of hormones and enzymes, and prevents obesity [54]. Considering an adult organism’s recommended daily calcium needs (1200 mg/kg), a sample with 25% BP can contribute significantly to meeting this requirement.

BP is also a rich source of phosphorus (2602 mg/kg, Table 6). However, considering the even larger amount of phosphorus in spelt flour (3763 mg/kg), a statistically significant decrease in phosphorus concentration was noticeable with increasing BP contents in the biscuit samples (3255–2750 mg/kg). This indicates that the phosphorus originated mainly from the spelt flour. Sulfur is involved in many bodily processes, including the metabolism of proteins, carbohydrates, and fats. It promotes circulation, digestion, and removal of harmful substances from the body, has a strong effect against microbes, suppresses inflammation, and balances blood pressure, cholesterol, and sugar [55]. Due to the rich sulfur content of spelt flour, the results of this study showed that the sulfur content in the biscuits decreased with increases in the content of beetroot powder (Table 6).

As shown in Table 7, the most dominant microelements in the analyzed samples were iron and zinc. The concentration of iron in the BP was 13.66 mg/kg. The iron content in the biscuit samples statistically significantly decreased with an increase in the proportion of BP (15.46–11.83 mg/kg). The reason for this is the high concentration of iron in spelt flour (23.32 mg/kg). A similar situation was observed for zinc. Spelt flour contains a higher amount of zinc (23.78 mg/kg) than BP (19.66 mg/kg), so as a result of increasing the proportion of BP (and decreasing the proportion of spelt flour), the zinc content in the samples decreased significantly (16.45–13.32 mg/kg).

Among the analyzed elements, Pd, As, and Co were below the method quantification limits, while the concentrations of Cr, Se, Al, Li, Cd, and Ni were below 0.9 mg/kg. The content of Mn in the biscuit samples ranged from 5.27 to 8.16 mg/kg, while the Cu content was between 2.20 and 2.64 mg/kg. In addition, the concentrations of B, Sr, and Ba were below 3.12 mg/kg in all the biscuit samples.

It should be noted that the observed deviations in mineral content, particularly in samples C and D, might result from differences in the bioavailability of the elements, interactions between ingredients, and potential matrix effects during biscuit formulation and baking. Additionally, variations in water content and ingredient distribution could have influenced the final mineral concentrations. The presented screening of valuable macro- and microelements demonstrated that the analyzed beetroot biscuits have the potential to provide a significant concentration of minerals in the human diet.

### 3.5. Acrylamide Content

Acrylamide (AA) is a low-molecular-weight organic compound that dissolves in water and is formed via Maillard reactions during thermal processing and food exposure to temperatures above 120 °C. Acrylamide is mainly found in baked or fried foods that are rich in carbohydrates, where it is formed in reactions between the amino acid asparagine and reducing sugars such as glucose, fructose, and lactose [56]. The AA content depends on the time and temperature of baking/frying, the amount of asparagine in the food, and the availability of sugar. Foods such as chips, French fries, coffee, biscuits, and bakery products contribute the most to the total intake of AA in the human body [56]. The European Food Safety Authority (EFSA) announced in 2015 that AA in food is a public health problem (EFSA CONTAM Panel, 2015). The EFSA warns that the content of AA in food potentially increases the risk of cancer in people of all age groups. Due to its harmful effects, it is necessary to reduce the presence of AA in food that contains its precursors in raw form. The reference value for acrylamide in biscuits is 350 μg/kg (Regulation on maximum concentrations of specific contaminants in food, Official Gazette of the Republic of Serbia, 2022). In this study, the influence of the BP content (15–25%) in the biscuits and the baking temperature (150 and 170 °C) on AA formation was investigated. The acrylamide content in all the analyzed samples was far below the reference value of 350 μg/kg (Table 8). In the control samples, the AA content was higher at both baking temperatures (48 and 76 μg/kg at 150 and 170 °C, respectively) compared to the AA values in the beetroot biscuits. In addition, a statistically significant difference (*p* < 0.05) of AA (36–50 μg/kg) was found in the biscuits prepared at a higher baking temperature compared to the samples prepared at a lower temperature (15–30 μg/kg). With the increase in the beetroot content, the AA concentration decreased at both baking temperatures. Given that the protein content in all analyzed samples was about 9%, and the proportion of glucose and fructose was very low [23], it can be concluded that the most likely reason for the decrease in AA concentration with increasing BP contents is the presence of a significant amount of antioxidants [57]. In general, it can be concluded that the AA content in the beetroot biscuits was low due to the relatively low baking temperatures and the presence of antioxidants in the beetroot. For the sake of comparison, in the recent paper by Žilić et al. [58], the level of acrylamide in cereal biscuits baked at 180 °C for 13 min ranged from 72.3 to 861.7 μg/kg, with 30% of the biscuits exceeding the reference value.

### 3.6. Sensory Evaluation

Table 9 shows the results of evaluating the sensory quality of the biscuits for two test periods: immediately after production and after 6 months of storage. The biscuits were stored at room temperature in closed PET containers. Eight trained assessors with training certificates according to the requirements of the ISO 8586:2023 standard took part in the sensory assessment. In the first period of testing, immediately after production, the samples baked at a higher temperature, with 20 and 15% BP: **C2** ($\bar{X}$_avr_ = 4.69) and **B2** ($\bar{X}$_avr_ = 4.52), had excellent overall quality. These samples were rated best in terms of texture–structure, bakedness, and breakage immediately after production, which was due to their preparation at a higher temperature. Sample **C2** also received excellent scores for the other evaluated parameters, which were exceptionally high for taste, smell, and appearance (Table 9). The above-mentioned properties were also rated highly for sample **B2** in the first testing period. According to the obtained scores, sample **B1** with 15% BP, baked at a lower temperature, was on the borderline between very good and excellent quality ($\bar{X}$_avr_ = 4.47). The rest of the examined fresh samples belonged to the very good quality category. A decrease in sensory quality was observed in all the samples during storage, which is common and expected during storage. The observed decline was not pronounced, and after 6 months of storage, all the tested samples had very good overall quality. Based on the obtained results, it was observed that the samples with 20% BP had the sweetest taste, while the sample with 25% BP had a slightly earthy beetroot taste. The color was spotty in places, probably due to uneven mixing during preparation of the dough. The samples baked at a lower temperature obtained better color scores. The texture of samples **A**, **B**, and **C** was defined as slightly crunchy, while sample **D** was defined as chewy. The smell of all the samples was characteristic of beetroot, more or less pronounced. The samples with 20% BP received the best odor rating. The baking temperature and storage time did not have a significant influence on the sensory quality parameters (Table 9). It can be seen that sample **C2** retained its sensory quality best even after six months of storage, as it received the highest overall score ($\bar{X}$_avr_ = 4.03) in the second evaluation period and was therefore the best-rated sample in this study. Although it was rated with the lowest mean score for overall fresh quality ($\bar{X}$_avr_ = 4.13), sample **D2** exhibited a slight decline in sensory quality during storage, as it was rated immediately after sample **C2** at the end of the test period ($\bar{X}$_avr_ = 4.01). The sensory properties of sample **B2** also did not change significantly during storage.

Samples **C1** and **D1** exhibited a very similar sensory quality profile, with almost identical scores for overall sensory quality during both periods. In general, it can be concluded that the biscuits showed an improvement in taste with the addition of BP, which can be attributed to the unique flavor of beetroot. The decrease in flavor scores with an increase in BP content is a result of the dominance of the earthy and woody taste characteristic of beetroot. In addition, the decrease in taste scores was, to some extent, due to the development of bitter notes caused by the high tannin content of BP. Similar results were obtained by Sahni et al. [6], where biscuits with 10% beetroot showed better sensory properties than biscuits with 25% beetroot powder.

### 3.7. Glycemic Index

The estimation of the glycemic index (GI) is an important parameter to better understand the physiological effects of foods with high carbohydrate contents. Although spelt (*Triticum spelta* L.) is considered particularly interesting from a nutritional point of view, studies show that the glycemic response to bakery products made from spelt flour is similar to that of products made from wheat flour [59]. 

Consuming whole grain spelt flour can have a preventive effect on diabetes and obesity, while white spelt flour has a higher GI, similar to that of white wheat flour. According to the Diabetes Council, wholemeal spelt flour has a slightly lower GI than wholemeal flour made from buckwheat, maize, and millet [60]. A high GI (over 70) and glycemic load (GL) (over 20) characterize most standard confectionery products. An in vivo study conducted as part of this research on healthy subjects of both sexes showed the effects of the addition of BP on the glycemic response. Glycemia was measured at 0, 15, 30, 45, 60, 90, and 120 min after consuming 25.0 g of glucose and ~42.0 g of biscuits containing 25.0 g of available carbohydrates. The oral glucose tolerance test (OGTT) curve and the area under the curve (AUC) for the fortified biscuits showed significantly improved tolerance compared to pure glucose. Figure 3a shows the postprandial glucose concentration-dependence curve (OGTT) after consuming 25.0 g of pure glucose and 42.0 g of biscuits with 20% BP containing 25.0 g of available carbohydrates. The blood glucose values after the consumption of the biscuits were significantly lower than the values measured after glucose consumption. The peak value was reached 30 min after glucose consumption and was almost 9 mM, followed by a sharp drop. A very slight increase in glucose concentration after biscuit consumption reached a maximum of about 6 mM at the 45th minute, after which it returned to the initial value.

The areas under the resulting curves are shown in Figure 3b. For the time dependence curve of blood glucose concentration after the consumption of 25.0 g of pure glucose, more than twice the area was observed. Based on the obtained surfaces, the GI value of the biscuits with BP was 49 ± 11, indicating that this confectionery product can be classified as a food with a lower GI. The GL was 16.5, which places the biscuits in the medium GL category. The results indicate that beetroot biscuits can be recommended as a healthy substitute for standard confectionery products for consumers who are careful about their diet. This replacement would contribute to the prevention of obesity and diabetes. 

## 4. Conclusions

Incorporating beetroot powder (BP) with high stability during storage at different baking temperatures into biscuits resulted in a nutritionally enriched product with enhanced durability and appealing sensorial properties. The potassium content in the BP-enriched biscuits increased significantly, and the total betaine content reached half of the recommended daily intake. Hardness increased with higher BP contents but decreased significantly over six months of storage. The acrylamide levels in the enriched biscuits were lower than in the control and remained well below the reference threshold. Sensory analysis showed that biscuits with 20% BP received the highest ratings for appearance, texture, and a pleasant taste and aroma, with sensory properties remaining well-preserved after six months. The best-rated biscuit had a lower glycemic index than the control and a medium glycemic load. Based on presented results, it can be concluded that BP can serve as a natural alternative to synthetic additives, enhancing both nutritional value and sensory appeal in confectionery products. Future research will focus on industrial-scale production, as well as commercial and marketing aspects. Given the widespread consumption of biscuits, enriching them with fruit- and vegetable-based flours could positively impact public health.

## Figures and Tables

**Figure 1 foods-14-00814-f001:**
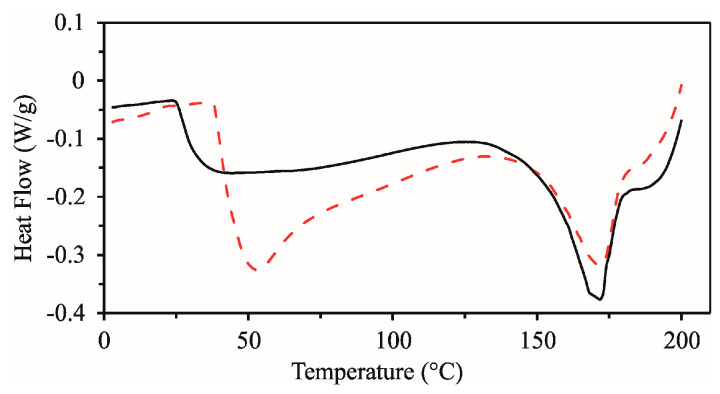
DSC curves of dehydrated beetroot sample at the beginning of storage (red dashed line) and after 6 months of storage at room temperature (solid line).

**Figure 2 foods-14-00814-f002:**
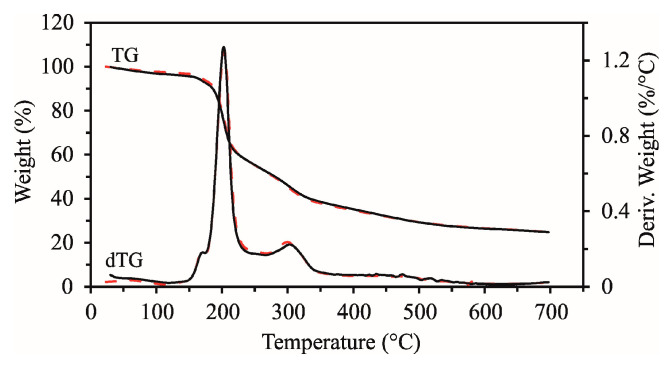
TG and dTG curves of a dehydrated beetroot sample at the beginning of storage (red dashed line) and after 6 months of storage (solid line).

**Figure 3 foods-14-00814-f003:**
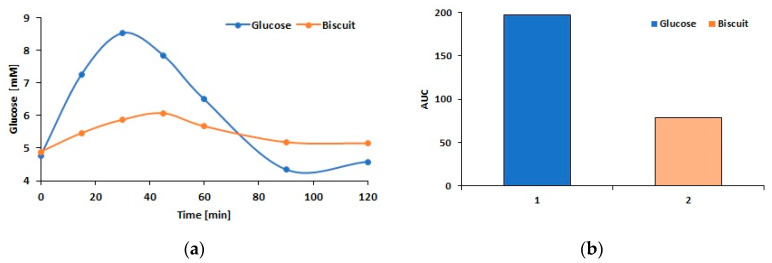
Postprandial glucose concentration dependence curve (OGTT) after consuming 25.0 g of pure glucose and an amount of biscuits with 20% BP containing 25.0 g of available carbohydrates (**a**); Areas under the time dependent curves of blood glucose concentration (AUC) after consumption of glucose and biscuits (**b**).

**Table 1 foods-14-00814-t001:** Designations of the biscuit samples and their corresponding BP (beetroot powder) content in the dough.

Sample Designation		Replaced Spelt Flour [%]	Mass Fraction of BP in the Dough [%]
150 °C	170 °C		
**A1**	**A2**	0	0
**B1**	**B2**	30	15
**C1**	**C2**	40	20
**D1**	**D2**	50	25

**Table 2 foods-14-00814-t002:** Thermodynamic parameters obtained from DSC curves of BP (beetroot powder) after six months of storage, as well as water activity and moisture content. Values are presented as mean ± SD.

Measurement		Start of Storage	6 Months
DSC	T_g,_ (°C)	41.00 ± 1.20	37.40 ± 2.00
	∆Cp (J/(g·°C))	3.01 ± 0.70	2.53 ± 0.80
	T_m_ (°C)	171.80 ± 1.60	171.70 ± 1.20
	∆ H (J/g)	38.78	43.70
Water activity (aw)		0.38 ± 0.001	0.36 ± 0.001
Moisture content (%)		6.80 ± 0.18	6.20 ± 0.20

**Table 3 foods-14-00814-t003:** Mass loss (%) and temperatures of different degradation stages obtained from TG and dTG curves of BP after dehydration and after 6 months of storage.

TGA	Start of Storage	6 Months
T_s_ 1 (°C)	29.0 ± 1.0	29.0 ± 2.0
T_end_ 1 (°C)	127.0 ± 1.0	127.0 ± 2.0
Mass loss 1 (%)	3.7 ± 0.5	3.0 ± 0.7
T_s_ 2 (°C)	127.0 ± 1.0	127.0 ± 2.0
T_end_ 2 (°C)	177.0 ± 2.0	174.0 ± 1.0
Mass loss 2 (%)	3.9 ± 0.3	3.5 ± 0.3
T_s_ 3 (°C)	177.0 ± 2.0	174.0 ± 1.0
T_end_ 3 (°C)	262.0 ± 1.0	269.0 ± 1.0
Mass loss 3 (%)	39.0 ± 3.0	42.0 ± 3.0
T_s_ 4 (°C)	262.0 ± 1.0	269.0 ± 1.0
T_end_ 4 (°C)	697.0 ± 1.0	698.0 ± 1.0
Mass loss 4 (%)	28.0 ± 1.0	27.0 ± 1.0
Total mass loss (%)	75.0 ± 1.0	75.0 ± 1.0
T_res_ (°C)	697.0 ± 1.0	698.0 ± 1.0
Residue (%)	25.0 ± 1.0	25.0 ± 1.0

**Table 4 foods-14-00814-t004:** The hardness [kg] of biscuits baked at different temperatures during storage. Significant differences in means (*p* < 0.05) within one type of biscuit between three time periods are shown with different superscript letters (a, b, c). Significant differences in mean values between four types of biscuits are shown by different superscript numbers (1, 2, 3, 4).

Sample	Start of Storage	3 Months	6 Months
**A1**	4.32 ± 0.80 ^1b^	4.15 ± 0.76 ^12a^	13.20 ± 1.24 ^4c^
**B1**	5.75 ± 2.04 ^1b^	3.94 ± 0.42 ^1a^	7.95 ± 1.45 ^2c^
**C1**	27.94 ± 0.74 ^3c^	7.15 ± 0.20 ^3a^	12.00 ± 0.03 ^3b^
**D1**	11.39 ± 1.01 ^2c^	4.48 ± 0.01 ^2a^	6.33 ± 0.88 ^1b^
**A2**	5.55 ± 0.46 ^1a^	6.88 ± 0.28 ^1b^	7.61 ± 1.39 ^1c^
**B2**	21.30 ± 4.28 ^2c^	15.33 ± 6.05 ^3b^	14.74 ± 6.33 ^2a^
**C2**	26.36 ± 3.38 ^3b^	9.02 ± 2.71 ^2a^	13.38 ± 4.87 ^2a^
**D2**	21.87 ± 6.11 ^3b^	11.45 ± 1.04 ^3a^	11.49 ± 1.97 ^2a^

**Table 5 foods-14-00814-t005:** Betaine content [mg/100 g] in biscuit samples after different storage periods. Significant differences in means (*p* < 0.05) within one type of biscuit between three time periods are shown with different superscript letters (a, b, c). Significant differences in mean values between four types of biscuits are shown by different superscript numbers (1, 2, 3, 4). SF—spelt flour, BP—beetroot powder.

Sample	Start of Storage	3 Months	6 Months
**SF**	131.32 ± 5.30 ^a^	141.49 ± 7.42 ^b^	128.25 ± 4.91 ^a^
**BP**	909.29 ± 14.46 ^b^	898.95 ± 34.77 ^a^	1197.32 ± 42.16 ^c^
**A1**	74.60 ± 2.20 ^1a^	72.20 ± 2.80 ^1a^	76.50 ± 2.11 ^1a^
**B1**	258.82 ± 11.19 ^2b^	217.08 ± 15.01 ^2a^	329.68 ± 18.16 ^2c^
**C1**	283.13 ± 18.17 ^2a^	279.58 ± 38.97 ^3a^	375.22 ± 7.15 ^3b^
**D1**	307.34 ± 5.25 ^3b^	292.29 ± 5.70 ^3a^	403.16 ± 9.16 ^4c^
**A2**	74.90 ± 3.55 ^1a^	77.90 ± 2.80 ^1a^	82.30 ± 3.14 ^1a^
**B2**	179.85 ± 6.90 ^2b^	135.91 ± 7.99 ^2a^	196.15 ± 8.12 ^2c^
**C2**	276.12 ± 0.28 ^3a^	266.68 ± 33.37 ^3a^	365.69 ± 38.16 ^3b^
**D2**	289.69 ± 9.20 ^4b^	274.64 ± 4.35 ^3a^	387.28 ± 7.12 ^3c^

**Table 6 foods-14-00814-t006:** Macroelement contents [mg/kg ± SD] in BP, SF, and biscuit samples prepared at 150 °C and 170 °C. Mean value with standard deviation is shown. Statistically significant differences (*p* < 0.05) between the mean values of composition components in the BP (beetroot powder), SF (spelt flour), and four types of biscuits are shown with different superscript letters (A, B, C, D).

Sample	Na	K	Mg	Ca	P	S
SF	10.45 ± 0.06	1410 ± 28.00	500 ± 13.19	127 ± 1.18	3763 ± 57.85	1271 ± 13.42
BP	2733 ± 10.45	7180 ± 144.13	580 ± 3.72	647 ± 18.83	2602 ± 10.23	908 ± 3.02
A1	564 ± 5.16 ^A^	1059 ± 9.63 ^A^	409 ± 5.63 ^A^	121 ± 4.46 ^A^	3255 ± 14.09 ^C^	780 ± 1.17 ^D^
B1	1077 ± 6.54 ^B^	2303 ± 24.28 ^B^	438 ± 6.30 ^C^	222 ± 4.59 ^B^	2750 ± 15.64 ^A^	667 ± 3.97 ^C^
C1	1225 ± 29.72 ^D^	2608 ± 14.97 ^C^	484 ± 5.14 ^D^	237 ± 6.24 ^C^	2760 ± 0.70 ^B^	646 ± 2.11 ^B^
D1	1196 ± 0.23 ^C^	2637 ± 68.42 ^C^	430 ± 7.99 ^B^	260 ± 5.76 ^D^	2750 ± 14.11 ^A^	622 ± 0.00 ^A^
A2	569 ± 5.19 ^A^	1074 ± 10.01 ^A^	424 ± 6.12 ^A^	125 ± 5.32 ^A^	3279 ± 15.03 ^D^	786 ± 1.21 ^C^
B2	1069 ± 9.41 ^B^	2283 ± 26.82 ^B^	436 ± 11.05 ^B^	275 ± 9.91 ^D^	2787 ± 10.58 ^B^	659 ± 1.88 ^B^
C2	1250 ± 1.86 ^D^	2660 ± 13.74 ^C^	478 ± 9.79 ^D^	238 ± 2.07 ^B^	2750 ± 6.75 ^A^	639 ± 2.56 ^A^
D2	1220 ± 15.51 ^C^	2743 ± 13.63 ^D^	470 ± 0.70 ^C^	270 ± 1.62 ^C^	2961 ± 0.94 ^C^	642 ± 2.35 ^A^

**Table 7 foods-14-00814-t007:** Content of microelements [mg/kg ± SD] in BP, SF, and BPB samples prepared at 150 °C and 170 °C. Mean values with standard deviations are shown. Statistically significant differences (*p* < 0.05) between the mean values of composition components in BP (beetroot powder), SF (spelt flour), and four types of biscuits are shown with different superscript letters (A, B, C, D).

Sample	Zn	Mn	Cu	Fe	Cr	Se	B	Al	Li	Ni	Sr	Ba	Cd
SF	23.78 ± 0.47	14.05 ± 0.32	3.53 ± 0.06	23.32 ± 0.63	<0.005	0.41 ± 0.06	0.21 ± 0.03	2.00 ± 0.09	0.07 ± 0.00	0.36 ± 0.00	0.31 ± 0.00	0.31 ± 0.00	0.02 ± 0.00
BP	19.66 ± 0.05	19.38 ± 3.72	4.73 ± 0.16	13.66 ± 0.94	<0.005	0.28 ± 0.02	12.94 ± 0.01	0.62 ± 0.15	0.92 ± 0.00	0.42 ± 0.00	8.61 ± 0.05	12.72 ± 0.04	0.20 ± 0.00
A1	16.45 ± 0.04 ^C^	7.84 ± 0.05 ^D^	2.28 ± 0.00 ^B^	15.46 ± 0.43 ^D^	0.06 ± 0.00 ^A^	0.60 ± 0.05 ^A^	<0.005	0.79 ± 0.10 ^D^	0.06 ± 0.00 ^A^	0.26 ± 0.00 ^B^	0.21 ± 0.00 ^A^	0.27 ± 0.03 ^A^	0.01 ± 0.00 ^A^
B1	13.25 ± 0.09 ^A^	6.85 ± 0.15 ^C^	2.15 ± 0.03 ^A^	12.78 ± 0.41 ^C^	0.40 ± 0.05 ^C^	0.68 ± 0.03 ^C^	1.96 ± 0.02 ^A^	0.50 ± 0.04 ^B^	0.14 ± 0.00 ^B^	0.33 ± 0.00 ^C^	1.45 ± 0.01 ^B^	2.15 ± 0.00 ^B^	0.02 ± 0.00 ^B^
C1	13.40 ± 0.03 ^B^	5.43 ± 0.07 ^A^	2.61 ± 0.02 ^C^	12.02 ± 0.01 ^B^	0.32 ± 0.04 ^B^	0.63 ± 0.00 ^B^	2.77 ± 0.01 ^B^	0.74 ± 0.18 ^C^	0.20 ± 0.00 ^D^	0.19 ± 0.00 ^A^	1.96 ± 0.02 ^D^	3.09 ± 0.07 ^C^	0.05 ± 0.00 ^C^
D1	13.32 ± 0.06 ^A^	6.61 ± 0.27 ^B^	2.24 ± 0.02 ^B^	11.83 ± 0.48 ^A^	<0.005	0.68 ± 0.00 ^C^	2.71 ± 0.01 ^B^	0.44 ± 0.09 ^A^	0.18 ± 0.00 ^C^	0.42 ± 0.00 ^D^	1.90 ± 0.06 ^C^	3.12 ± 0.07 ^C^	0.02 ± 0.00 ^B^
A2	16.56 ± 0.02 ^D^	8.16 ± 0.05 ^C^	2.33 ± 0.00 ^B^	15.98 ± 0.42 ^D^	0.26 ± 0.00 ^C^	0.68 ± 0.04 ^C^	<0.005	0.81 ± 0.11 ^C^	0.06 ± 0.00 ^A^	0.25 ± 0.00 ^B^	0.22 ± 0.00 ^A^	0.26 ± 0.02 ^A^	0.01 ± 0.00 ^A^
B2	14.77 ± 0.11 ^C^	6.54 ± 0.04 ^B^	2.20 ± 0.00 ^A^	14.67 ± 0.48 ^C^	<0.005	0.57 ± 0.00 ^A^	1.94 ± 0.01 ^A^	0.79 ± 0.02 ^B^	0.17 ± 0.00 ^B^	0.36 ± 0.00 ^C^	1.43 ± 0.00 ^B^	2.11 ± 0.03 ^B^	0.02 ± 0.00 ^B^
C2	13.51 ± 0.03 ^A^	5.27 ± 0.05 ^C^	2.64 ± 0.01 ^C^	11.69 ± 0.44 ^A^	0.25 ± 0.08 ^B^	0.69 ± 0.00 ^C^	2.71 ± 0.00 ^C^	0.90 ± 0.08 ^D^	0.20 ± 0.00 ^D^	0.20 ± 0.00 ^A^	1.88 ± 0.00 ^C^	2.99 ± 0.01 ^C^	0.05 ± 0.00 ^C^
D2	13.85 ± 0.03 ^B^	7.81 ± 0.10 ^C^	2.32 ± 0.00 ^B^	13.33 ± 0.21 ^B^	0.15 ± 0.02 ^A^	0.66 ± 0.05 ^B^	2.65 ± 0.02 ^B^	0.77 ± 0.06 ^A^	0.18 ± 0.00 ^C^	0.42 ± 0.00 ^D^	1.95 ± 0.01 ^D^	3.12 ± 0.03 ^D^	0.02 ± 0.00 ^B^

Concentrations of Pb, As, and Co were below limit of quantification (<0.005) in all analyzed samples.

**Table 8 foods-14-00814-t008:** Acrylamide concentration [µg/kg ± SD] in biscuits prepared at 150 and 170 °C. Significant differences in mean values (*p* < 0.05) between four types of biscuits are shown with different superscript letters (A, B, C).

Sample	A1	B1	C1	D1	A2	B2	C2	D2
AA	48.0 ± 7.1 ^C^	30.0 ± 5.6 ^B^	27.8 ± 7.2 ^AB^	15.4 ± 6.9 ^A^	76.3 ± 8.5 ^C^	50.0 ± 6.3 ^B^	38.1 ± 4.4 ^A^	35.6 ± 9.0 ^A^

**Table 9 foods-14-00814-t009:** Results of sensory evaluation of beetroot biscuits during storage.

Sample	Storage Period	Calculated Indicators	Color, Surface Size and Shape (Mean ± SD)	Structure, Bakedness, and Breakage (Mean ± SD)	Chewiness (Mean ± SD)	Smell (Mean ± SD)	Taste (Mean ± SD)	Overall Mean Score	% of Maximum Possible Quality
**A1**	fresh	$\bar{X}$ _avr_	4.40 ± 0.38	4.09 ± 0.13	4.25 ± 0.30	4.50 ± 0.42	4.16 ± 0.32	4.27	85.36
	after 6 months	$\bar{X}$ _avr_	3.87 ± 0.57	3.87 ± 0.53	3.81 ± 0.22	3.91 ± 0.42	3.12 ± 0.13	3.68	73.57
**B1**	fresh	$\bar{X}$ _avr_	4.7 ± 0.19	4.16 ± 0.26	4.16 ± 0.26	4.62 ± 0.23	4.59 ± 0.35	4.47	89.32
	after 6 months	$\bar{X}$ _avr_	4.22 ± 0.21	3.78 ± 0.36	3.84 ± 0.38	3.81 ± 0.61	3.37 ± 0.46	3.76	75.23
**C1**	fresh	$\bar{X}$ _avr_	4.84 ± 0.19	4.34 ± 0.19	4.22 ± 0.21	4.53 ± 0.09	4.53 ± 0.36	4.33	86.53
	after 6 months	$\bar{X}$ _avr_	4,22 ± 0.34	3.84 ± 0.26	3.91 ± 0.23	4.00 ± 0.30	3.72 ± 0.28	3.91	78.26
**D1**	fresh	$\bar{X}$ _avr_	4.37 ± 0.35	4.25 ± 0.27	4.06 ± 0.22	4.47 ± 0.16	4.28 ± 0.28	4.28	85.64
	after 6 months	$\bar{X}$ _avr_	3.69 ± 0.46	3.62 ± 0.30	3.69 ± 0.22	3.81 ± 0.42	3.62 ± 0.35	3.68	73.65
**A2**	fresh	$\bar{X}$ _avr_	4.44 ± 0.37	3.90 ± 0.19	3.97 ± 0.16	4.37 ± 0.23	4.09 ± 0.13	4.14	82.73
	after 6 months	$\bar{X}$ _avr_	4.16 ± 0.19	3.97 ± 0.36	3.91 ± 0.19	3.66 ± 0.32	3.28 ± 0.39	3.75	75.04
**B2**	fresh	$\bar{X}$ _avr_	4.72 ± 0.25	4.37 ± 0.19	4.19 ± 0.18	4.69 ± 0.26	4.66 ± 0.23	4.52	90.46
	after 6 months	$\bar{X}$ _avr_	4.22 ± 0.25	3.94 ± 0.32	3.87 ± 0.27	3.91 ± 0.42	3.69 ± 0.39	3.90	77.99
**C2**	fresh	$\bar{X}$ _avr_	4.75 ± 0.33	4.56 ± 0.18	4.56 ± 0.18	4.69 ± 0.26	4.87 ± 0.13	4.69	93.84
	after 6 months	$\bar{X}$ _avr_	4.28 ± 0.16	4.00 ± 0.19	4.00 ± 0.27	4.06 ± 0.29	3.91 ± 0.30	4.03	80.63
**D2**	fresh	$\bar{X}$ _avr_	4.25 ± 0.40	4.00 ± 0.13	3.72 ± 0.16	4.41 ± 0.23	4.25 ± 0.00	4.13	82.52
	after 6 months	$\bar{X}$ _avr_	4.12 ± 0.40	3.94 ± 0.26	4.00 ± 0.27	4.10 ± 0.23	3.94 ± 0.42	4.01	80.22

## Data Availability

The original contributions presented in this study are included in the article. Further inquiries can be directed to the corresponding authors.

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
