# Peer review of "Low Glycemic Index Biscuits Enriched with Beetroot Powder as a Source of Betaine and Mineral Nutrients"

_foods, 2025, doi:10.3390/foods14050814_

Round 1

Reviewer 1 Report

Comments and Suggestions for Authors

The number of independent repetitions of each analysis should be given either in the methodology or below the figures/tables.

lines 40/41: Beta vulgaris L.

section 2.1. The article should be as independent as possible and contain the most important information; this is not the second part of the article by Mitrevski et al. Since beetroot powder is one of the main materials studied, its origin - the method of obtaining it, the number of independent repetitions (batches), and a short characteristic should be given.

Also give the number of independent repetitions, i.e. how many batches of biscuits were baked according to each recipe.

line 122: beetroot powder or flour ? be consistent

lines 121/122: Three independent samples from 3 different batches of beetroot powder ? If 3 samples were taken from one batch, they are not independent samples (please compare the note to section 2.1.).

lines 148/149: why don't you express hardness in N ?

line 155: extraction efficiency depends, among other things, on the degree of material fragmentation, were all samples sieved after being grounded to unify the particle size in the tested materials ?

section 2.11. The study has many factors - BP content, baking temperature, and storage time. The tables show that the various parameters were analyzed separately, for example, one group was samples baked at 150 oC, the other at 170 oC. Why wasn't a two- or three-way analysis of variance used to determine the significance of the effect of individual factors ?

lines 412-416: How can this increase be explained ?

section 3.4. If we replace an ingredient with another with different content of the macro- and microelements, their content in biscuits should result from the amount of both ingredients and the content of the tested macro- and microelements in them. Meanwhile, Tables 6 and 7 show large disruptions in the trend that should be expected, especially in samples C and D. How can this be explained ?

line 590: Triticum spelta L.

Tables 2, 3, and 9: statistical significance of the differences was not tested or all values at 0 and 6 months were not significantly different ?

Figure 2: Mark which lines indicate TG and which lines indicate dTG. Since the solid and dashed lines overlap perfectly, you can't tell them apart (in which case it doesn't really matter, but it may be worth marking some of them with a different color to make it more visible for the reader's convenience).

Table 9: The table description is a bit chaotic, where are the importance coefficients?

Abbreviations (such as BP, SF, BPB, etc.) should be explained under each table/figure.

Figure 3: Comparison with pure glucose yielded obvious and predictable results. The proper reference for biscuits enriched with the beetroot powder should be biscuits obtained from the basic recipe (A1, A2). Did the baking temperature make any difference here ?

Author Response

We sincerely thank the reviewer for their constructive suggestions, which have helped improve the quality of our manuscript.

Below, we provide a point-by-point response detailing how each suggestion has been addressed. All revisions in the manuscript are highlighted in yellow.

Reviewer#1

Q. The number of independent repetitions of each analysis should be given either in the methodology or below the figures/tables.

A. Thank you for your valuable suggestion. The number of independent repetitions of the analysis has been included in the methodology section.

Q. lines 40/41: Beta vulgaris L.

A. It is corrected in the revised version of the manuscript. 

Q. Section 2.1. The article should be as independent as possible and contain the most important information; this is not the second part of the article by Mitrevski et al. Since beetroot powder is one of the main materials studied, its origin - the method of obtaining it, the number of independent repetitions (batches), and a short characteristic should be given.

A. Thank you for your valuable comment. The following text is added in the revised version of the manuscript.

The beetroot of the Detroit variety was washed thoroughly and the skins were peeled, then cut into 1 mm thick rings. The rings were placed on drying trays without overlapping and baked at 52 °C for 24 hours to obtain a constant mass of dehydrated beetroot. After cooling for 3 hours, the dried rings were ground in a blender specially designed for grinding grains (VITA-MIX CORP, 1200W, USA).

The biscuits were prepared by replacing a certain percentage of the whole meal spelt flour (SF) with beetroot powder (BP). The biscuits were prepared in two batches and baked at 150 °C and 170 °C. A blank test without beetroot powder was prepared for each batch. All biscuits were baked in a multi-stage oven (each type of biscuit was on its own separate tray, in sum 20 biscuits on one tray). After baking, the biscuits were left on the baking tray for 2 hours to cool. The ingredients are listed in Table 1.

Q. Also give the number of independent repetitions, i.e. how many batches of biscuits were baked according to each recipe.

A. Thank you for your kind comment. It has been included in the revised version of the manuscript. Please see the text in the previous comment. 

Q. line 122: beetroot powder or flour? be consistent

A. Thank you for your valuable comment. It should be beetroot powder and accordingly the correction has been made throughout the manuscript. 

Q. lines 121/122: Three independent samples from 3 different batches of beetroot powder? If 3 samples were taken from one batch, they are not independent samples (please compare the note to section 2.1.).

A. Thank you for your observation. It has been corrected in the revised version of the manuscript. 

Q. lines 148/149: why don't you express hardness in N?

A. Thank you for your valuable comment. While Newtons (N) are the SI unit, we presented hardness in kilograms (kg) as it is commonly used in texture analysis of biscuits and aligns with previous studies for better comparability.

Q. line 155: extraction efficiency depends, among other things, on the degree of material fragmentation, were all samples sieved after being grounded to unify the particle size in the tested materials?

A. Thank you for your comment. The entire extraction method has been optimized in terms of the amount of extraction solvent, duration of ultrasonic treatment, vortexing time, etc. Regarding particle size, the material was previously ground into fine particles to ensure uniformity and achieve a finely powdered sample. The method has been fully optimized, validated, and published in the following study:

Kojić, J.; Krulj, J.; Ilić, N.; Lončar, E.; Pezo, L.; Mandić, A.; Bodroža Solarov, M. Analysis of Betaine Levels in Cereals, Pseudocereals and Their Products. J Funct Foods 2017, 37, 157–163. doi:10.1016/j.jff.2017.07.052.

Q. Section 2.11. The study has many factors - BP content, baking temperature, and storage time. The tables show that the various parameters were analyzed separately, for example, one group was samples baked at 150 oC, the other at 170 oC. Why wasn't a two- or three-way analysis of variance used to determine the significance of the effect of individual factors?

A. Thank you for your valuable comment. The structure of the analysed data is such (there was no normal distribution) that we could not apply parametric statistics such as analysis of variance. We therefore used non-parametric tests (Kruskal-Wallis and Wilcoxon-Mann-Whitney test). Although there are non-parametric alternatives to two-way or three-way ANOVA (e.g. the Friedman test), as far as we know from the literature, there is no complete agreement on the analogy between these two methods. Therefore, we decided to use the above tests when analysing our data. If you think that the results could be significantly different by using other methods, we can also perform analyses and tests that you recommend for our data as valid.

Q. lines 412-416: How can this increase be explained?

A. Thank you for your kind question. The increase in betaine content over six months of storage is likely due to moisture loss, which concentrates water-soluble compounds like betaine, and structural changes in the biscuit matrix, such as starch retrogradation or protein modifications, which enhance betaine's extractability (Korus et al., 2015; Nayak et al., 2014). Additionally, protein interactions with betaine may weaken over time, leading to higher measured levels (de Zwart et al., 2003). Storage conditions, including humidity and temperature, could further influence this trend.

References:

Korus, J., Witczak, T., Ziobro, R., & Juszczak, L. (2015). The influence of acorn flour on rheological properties of gluten-free dough and physical characteristics of the bread. Eur Food Res Technol, 240, 1135–1143.

Nayak, B., Liu, R. H., & Tang, J. (2014). Effect of processing on phenolic antioxidants of fruits, vegetables, and grains—a review. Crit Rev Food Sci Nutr, 55(7), 887-920.

de Zwart, F. J., Slow, S., Payne, R. J., Lever, M., George, P. M., Gerrard, J. A., & Chambers, S. T. (2003). Glycine betaine and proline betaine in human blood and urine. Biochim Biophys Acta, 1637(2), 115-120.

Q. Section 3.4. If we replace an ingredient with another with different content of the macro- and microelements, their content in biscuits should result from the amount of both ingredients and the content of the tested macro- and microelements in them. Meanwhile, Tables 6 and 7 show large disruptions in the trend that should be expected, especially in samples C and D. How can this be explained?

A. Thank you for your insightful comment. The observed deviations in the macro- and microelement trends, particularly in samples C and D, have been acknowledged. These variations can be attributed to differences in the bioavailability and interaction of minerals during biscuit formulation and baking. Additionally, factors such as ingredient distribution, water content changes, and possible matrix effects might have influenced the final element composition. To clarify this, a brief explanation has been added in Section 3.4."

Q. line 590: Triticum spelta L.

A. It is corrected in the revised version of the manuscript.

Q. Tables 2, 3, and 9: statistical significance of the differences was not tested or all values at 0 and 6 months were not significantly different?

A. Thank you for your valuable comment. The values at beginning of the storage and after 6 months were not significantly different.  

Q. Figure 2: Mark which lines indicate TG and which lines indicate dTG. Since the solid and dashed lines overlap perfectly, you can't tell them apart (in which case it doesn't really matter, but it may be worth marking some of them with a different color to make it more visible for the reader's convenience).

A. Thank you for your valuable comment.

Figure 2: The TG and DTG curves have been marked. For the reader’s convenience, the black dashed line has been changed to red in Figures 1 and 2.

According to the reviewer’s suggestion, the legend under Figures 1 and 2 has been changed to:

Figure 1. DSC curves of a dehydrated beetroot sample, at the beginning of storage (red dashed line) and after 6 months of storage at room temperature (solid line).

Figure 2. TG and dTG curves of a dehydrated beetroot sample, at the beginning of storage (red dashed line) and after 6 months of storage (solid line).

Q. Table 9: The table description is a bit chaotic, where are the importance coefficients?

A. Thank you for your constructive suggestion. Table 9 has been modified for better clarity.

Q. Abbreviations (such as BP, SF, BPB, etc.) should be explained under each table/figure.

A. Thank you for your suggestion. It is included in the revised version of the manuscript. 

Q. Figure 3: Comparison with pure glucose yielded obvious and predictable results. The proper reference for biscuits enriched with the beetroot powder should be biscuits obtained from the basic recipe (A1, A2). Did the baking temperature make any difference here?

A. Thank you for your valuable comment. The baking temperature did not have a significant impact on the results.

Reviewer 2 Report

Comments and Suggestions for Authors

Beetroot is an important raw material of plant origin with proven positive effects on the human body. Red beetroot is a rich source of minerals, phenol compounds, carotenoids, glycine betaine, saponins, polyphenols and flavonoids.

Foods with high nutritional value are in great demand for proper functioning of body systems and potential health benefits. The incorporation of composite flour into traditional wheat based food products provided additional nutrients from non-wheat material and improved the nutritional value of the products.

The bakery industry is one of the largest organized food industries all over the world and in particular biscuits is one of the most popular products because of their convenience, ready to eat nature, and long shelf life. Recently, increasing consumer demand for healthier foods has triggered the development of biscuits made with natural ingredients exhibiting functional properties and providing specific health benefits beyond those to be gained from traditional nutrients.

The glycemic index concept was proposed in order to give complementary information about the chemical composition of carbohydrate-rich foods. The intake of low glycemic index foods may benefit weight regulation and be favourable  in the treatment of obesity, and is an important factor for maintaining a health diet.

The present study investigated the effect of replacing spelt flour with beetroot powder on various functional properties of biscuits, including betaine content, macro- and microelements, glycemic index, thermal analysis, the acrylamide content, the toxic elements (Pb, As, Co) and sensory analysis.

Reccomendations

The paper is clear and able to read by the focus groups. The conclusions are well present. Proposed experimental conditions and results could gain high correlated and it is reliable.

L 112 Please present briefly the stages of the technological process and the materials used

Accept with minor revisions

Author Response

We sincerely thank the reviewer for their constructive suggestions, which have helped improve the quality of our manuscript.

Below, we provide a response detailing how each suggestion has been addressed. All revisions in the manuscript are highlighted in yellow.

The paper is clear and able to read by the focus groups. The conclusions are well present. Proposed experimental conditions and results could gain high correlated and it is reliable.

A. Thank you very much for your supportive and encouraging comment.

Q. L 112 Please present briefly the stages of the technological process and the materials used.

A. Thank you for your valuable comment. The suggested information has been incorporated in the revised vision of the manuscript.

Reviewer 3 Report

Comments and Suggestions for Authors

In this manuscript entitled "Low glycemic index biscuits enriched with beetroot powder as a source of betaine and mineral nutrients", the authors investigated the effect of replacing spelt flour with beetroot powder (BP) on various functional properties of biscuits (BPB), including betaine content, macro- and microelements, and glycemic index and load. BP could be considered a functional ingredient that efficiently replaces synthetic additives in confectionery products such as biscuits and other foods. This manuscript is innovative and practical, rich in data and reasonable in the analysis of results, but needs further improvement in the introduction, materials and methods and reference format. In conclusion, I think it can  solving the following problems.

1. The abstract needs to be supplemented with the purpose of this study.

2. The main chemical components and their proportions of beetroot powder need to be supplemented in the introduction.

3. What is the difference between beetroot powder and flour used to make cookies? Why can beetroot powder replace some flour? These need to be compared and summarized in the preface.

4. The main purpose and innovative points of this study need to be condensed in the preface.

5. Why were 150 ºC and 170 ºC chosen as the baking parameters for cookies? Does high temperature have an impact on the content of betaine?

6. What type of probe was used to measure the hardness of the biscuits in 2.5? This needs to be completed.

7. Are the extraction and measurement methods of betaine in 2.6 supported by references? Please confirm.

8. What is the age distribution of people with hyperglycemia? What is the specific number of volunteers at different ages in 2.10?

9. Biscuit sample storage conditions and storage days need to be supplemented in the materials and methods.

10. The number of repetitions of the experiment should be added in 2.11.

11. What samples are used for DSC and TGA measurements? Is it dough with BP in it?

12. In Table 4, the hardness values of different types of cookies are significantly different. It is suggested to change to lowercase letter abc. Similarly, Table 5 needs to be modified.

13.  What is the full name of SF in Table 5? Please add comments.

14. The conclusion needs to be concise.

15. The reference format needs to be re-proofread to be consistent with the requirements of this journal.

Author Response

We sincerely thank the reviewer for their constructive suggestions, which have helped improve the quality of our manuscript.

Below, we provide a point-by-point response detailing how each suggestion has been addressed. All revisions in the manuscript are highlighted in yellow.

Q. The abstract needs to be supplemented with the purpose of this study.

A. Thank you for your constructive suggestion. The abstract has been revised to clearly state the purpose of the study.

Q. The main chemical components and their proportions of beetroot powder need to be supplemented in the introduction.

A. Thank you for your valuable comment. As suggested, the Introduction has been supplemented with additional information on the main chemical components and their proportions in beetroot powder.

Q. What is the difference between beetroot powder and flour used to make cookies? Why can beetroot powder replace some flour? These need to be compared and summarized in the preface.

A. Thank you for your observation.

Spelt flour is made by grinding whole spelt grains, and beetroot powder is made by drying and grinding raw beetroot. The idea is to replace part of the spelt flour with beetroot powder in order to improve the nutritional value of the final product. In order not to confuse readers, the term ‘beetroot flour’ is replaced by ‘beetroot powder’ throughout the manuscript.

Q. The main purpose and innovative points of this study need to be condensed in the preface.

A. We appreciate your insightful feedback and have revised the text accordingly to better highlight the main purpose and innovative aspects of our study.

Q. Why were 150 °C and 170 °C chosen as the baking parameters for cookies? Does high temperature have an impact on the content of betaine?

A. Thank you for your valuable comment. Bearing in mind that these are biscuits with the addition of beets as an active ingredient and that the goal was to obtain product in which the bioactive properties will be preserved as much as possible while achieving the desired sensory attributes, preliminary tests related to the baking mode were carried out. Initially, higher baking temperatures were used, but it was found that these did not reflect well on the sensory acceptability of the biscuits. For this reason, milder regimes were chosen, with a temperature difference of 20 °C, which was considered to be sufficient to observe the changes in biscuit properties originating from the baking temperature.

The influence of baking temperature on the betaine content has given in the manuscript (lines 435-438).

Q. What type of probe was used to measure the hardness of the biscuits in 2.5? This needs to be completed.

A. Thank you for your valuable comment. The measurements were conducted following the method described by Belović et al. (Potential Application of Triticale Cultivar “Odisej” for the Production of Cookies, Ratarstvo i povrtarstvo 2020, 57, 8–13, doi:10.5937/ratpov57-24126). According to this method, the instrument was equipped with a flat blade (HDP/BS) and a 30 kg load cell, as cited in the referenced study. This approach ensures methodological consistency and comparability with previous research.

Q. Are the extraction and measurement methods of betaine in 2.6 supported by references? Please confirm.

A. Thank you for your valuable comment. The proper reference has been added in part 2.6. of the manuscript.  

Q. What is the age distribution of people with hyperglycemia? What is the specific number of volunteers at different ages in 2.10?

A. Thank you for your valuable comment. The specific number of volunteers, the body mass index and fasting glucose level is included in the manuscript. 

Q. Biscuit sample storage conditions and storage days need to be supplemented in the materials and methods.

A. Thank you for your valuable comment. It is included in the revised version of the manuscript. 

Q. The number of repetitions of the experiment should be added in 2.11.

A. Thank you for your valuable comment. An explanation of the number of replicants has been added in the revised version of the manuscript.

Q. What samples are used for DSC and TGA measurements? Is it dough with BP in it?

A. Thank you for your comment. Dehydrated beetroot sample (BP powder) was used for DSC and TGA measurements.

Q. In Table 4, the hardness values of different types of cookies are significantly different. It is suggested to change to lowercase letter abc. Similarly, Table 5 needs to be modified.

A. Thank you for your valuable comment. It has been corrected as suggested. 

Q. What is the full name of SF in Table 5? Please add comments.

A. Thank you for your comment. The Abbreviations in all tables are defined in the revised version of the manuscript. 

Q. The conclusion needs to be concise.

A. Thank you for your valuable comment. The conclusion has been revised to be more concise.

Q. The reference format needs to be re-proofread to be consistent with the requirements of this journal.

A. Thank you for your observation. The Mendeley program was used to format the references in the manuscript. However, some errors were found which will be corrected in the pre-proofreading version.

Round 2

Reviewer 3 Report

Comments and Suggestions for Authors

All issues were addressed.

Author Response

Dear Editor,

Thank you for your comment regarding the way significant differences in the mean values of different parameters are marked in the tables. The essence of labelling in the way we have used is that we are trying to present a large number of results obtained in this extensive analysis with a smaller number of tables. The problem with this type of presentation is that it can be complicated to follow given the large amount of data in the tables created in this way. Labelling significant differences with different letters (a, b, c and d) shows that the mean values with different letters are statistically significantly different. However, it was not our intention to use the letters to indicate which mean values are higher or lower. As you can see in the part of the text that refers to the results of e.g. Table 5, the letters a, b and c are not mentioned, but it is described in detail which mean values are higher and which are lower.

In the same way, we have used the numbers 1, 2, 3 and 4 to indicate significant differences in the tables.

However, as the significance of the differences presented in this way could lead to a dilemma for future readers of our journal, we have changed the labels in the tables to make them intuitively clearer even without a detailed comparison with the text for this table. With this in mind, we have changed the labelling in Tables 4, 5, 6, 7 and 8 in the way you have suggested.